# Element-specific X-Ray detection of electron paramagnetic resonance in thin films of quantum bits

Andrin Doll [1,2] ✉, Zhewen Xu[1], Vladyslav Romankov[1], Giovanni Boero [3], Stefano Rusponi [4], Harald Brune [4], Zaher Salman [2] & Jan Dreiser [1] ✉

Element-specific magnetism accessible by synchrotron-based X-ray spectroscopy has proven to be valuable to study spin and orbital moments of transition metals and lanthanides in technologically relevant thin-film and monolayer samples. The access to coherent spin superposition states relevant for emergent quantum technologies remains, however, elusive with ordinary X-ray spectroscopy. Here, we approach the study of such quantum-coherent states via the X-ray detection of microwave-driven electron paramagnetic resonance, which involves much smaller signal levels than X-ray detected ferromagnetic resonance on classical magnets. We demonstrate the feasibility of this approach with thin films of phthalocyanine-based metal complexes containing copper or vanadium centers. We also identify X-ray specific phenomena that we relate to charge trapping of secondary electrons resulting from the decay of the X-ray excited core-hole state. Our findings pave the way toward the element-specific X-ray detection of coherent superposition states in monolayers of atomic and molecular spins on virtually arbitrary surfaces.

Recently, interest in versatile, coherent quantum systems has grown massively due to the rise of quantum technologies[1]. Molecular qubits are of small, nanometer size, and they can be functionalized and easily organized on surfaces[2]. Reports on Rabi oscillations[3–6] and room-temperature coherence[7] in molecular qubits have demonstrated the suitability and competitiveness of these molecular materials. Progress in this field does, however, critically depend on the availability of matching characterization techniques. Conventional electron paramagnetic resonance (EPR) is, in principle, able to measure coherent spin dynamics in thin films[8]. Monolayers of molecular qubits employing metal ions as spin-carrying units are, however, below the detection threshold of the ordinary EPR and would require emerging quantum sensing techniques[9,10]. Scanning probe techniques can access the quantum state of single spins or molecules on surfaces, as demonstrated by magnetic force microscopy[11], scanning tunneling microscopy[12], or atomic force microscopy[13]. Moreover, certain specific

molecules allow for single-spin readout via optical fluorescence[14]. For more general optical spin readout techniques such as Faraday rotation[15] and magnetic circular dichroism[16], however, sensitivity is limited to bulk quantities.

X-ray magnetic circular dichroism (XMCD) allows to detect the magnetic moment of a given atomic species with ultrahigh sensitivity. In the total electron yield mode, the detection threshold is well below a single monolayer of materials[17,18]. So far, this technique has been widely used to investigate, among others, molecular magnets, single-molecule magnets, and single-atom magnets on surfaces and to study the influence of the interface on the magnetic properties[19–21]. EPR, by contrast, allows the manipulation of the magnetic moments by resonant excitation. Spectrally, EPR can be extremely selective due to the very narrow line widths allowing to separate out different hyperfine spin states. It is thus beneficial to combine the element specificity and high sensitivity of XMCD with the ultrahigh spectral resolution of

[1]PSI Center for Photon Sciences CPS, Villigen PSI, Switzerland. [2]PSI Center for Neutron and Muon Sciences CNM, Villigen PSI, Switzerland. [3]Microsystems Laboratory, École Polytechnique Fédérale de Lausanne, Lausanne, Switzerland. [4]Institute of Physics, École Polytechnique Fédérale de Lausanne, Lausanne, Switzerland. ✉e-mail: andrin.doll@psi.ch; jan.dreiser@psi.ch

EPR. Generally, X-ray detected EPR (XDEPR) requires a detection scheme similar to the already demonstrated X-ray detected ferromagnetic resonance (XFMR)[22–25]. XFMR works with classical magnets; however, to access the quantum properties of isolated spins through XDEPR, a significant, non-incremental improvement of the detection scheme is required because the signal levels are several orders of magnitude lower compared to XFMR.

Here, we deliver a proof-of-concept demonstration of XDEPR, revealing that it is possible to detect microwave-driven changes in $Cu^{2+}$ and $V^{4+}$ ions' longitudinal magnetic moments using XMCD. Copper-phthalocyanine (CuPc) and vanadyl-oxophthalocyanine (VOPc) are ideal candidates for these experiments because these molecules are chemically stable[26], they exhibit long spin coherence times[8], and high-quality films with variable dilution levels can be easily grown[26,27].

## Results

### Magnetic signatures in the X-ray and EPR spectra

Figure 1a, c shows X-ray absorption and XMCD spectra recorded on diluted films of CuPc and VOPc at the copper and vanadium $L_{2,3}$ edges in transmission mode, respectively, at a temperature of ~6 K and an applied magnetic field of 3 T. The thin-film samples of CuPc and VOPc (thicknesses of 1 μm and 0.5 μm, respectively) diluted in compatible diamagnetic host materials were grown on thin X-ray transparent aluminum foils. These samples denoted as Cu-1 and V-1 in Supplementary Table 2, were used to obtain all presented experimental results unless stated otherwise. The molecular structures of CuPc and VOPc featuring the central metal ions and the phthalocyanine macrocycles are illustrated in Fig. 1a, c. The X-ray spectra clearly show the signatures of the transitions of the 2p → 3d atomic shells. The groups of features belonging to the $L_2$ and $L_3$ edges are separated due to the spin-orbit coupling in the Cu and V 2p shells[28–30]. The VOPc spectrum exhibits a lot of features due to additional multiplet effects. In contrast, the CuPc spectrum only shows two main peaks because of the closed-shell $3d^{10}$ final state[31] as opposed to the $3d^2$ final state for VOPc. Figure 1b displays a pulsed EPR spectrum acquired at a fixed microwave frequency of $f = 6$ GHz at ~5 K, recorded on the same diluted CuPc sample used to obtain the X-ray spectra shown in Fig. 1a (see also Supplementary Note 7 in the SI). In agreement with the simulation obtained using literature parameters[8,32], the experimental spectrum reveals four peaks related to the copper nuclear spin of $I_{Cu} = 3/2$. The shape is characteristic of oriented CuPc molecular films as reported in the literature[8]. At a magnetic field of ~220 mT corresponding to the free electron g-factor $g_{free} = 2.002$, which is indicated by the dashed

green line, a slightly stronger feature related to a deliberately introduced spectroscopic reference material is visible, namely the Koelsch radical. This reference was placed on top of the CuPc sample to calibrate pulse rotation angles for pulsed EPR (see Supplementary Note 7 in the SI), whereas all X-ray data were acquired prior to the insertion of this reference.

### Observation of X-ray detected EPR

We now turn to the X-ray detected EPR (XDEPR) experiments. A sketch of the core experimental setup is shown in Fig. 2a. The molecular film sample (blue) was placed in the X-ray transmission pathway. While sending the X-ray beam through, a half-wave microwave resonator (orange) working at a frequency of $f = 7.5$ GHz allowed the application of an oscillating magnetic field $B_1$ in the sample plane (orange arrow). The static external field $B_{ext}$ oriented perpendicular to the sample plane is shown as a red arrow. The transmitted X-ray intensity was then detected by a photodiode. The microwave amplitude was modulated at a frequency $\omega_{mod}/2\pi$ of a few hundred Hz, and a lock-in detection was employed to reduce the noise floor. With the microwave frequency fixed by the resonator geometry, XDEPR spectra were acquired by sweeping the magnetic field $B_{ext}$ and by simultaneously recording the intensity of the transmitted circularly polarized X-rays. In the XDEPR experiments, which will be described below, the spin polarization was monitored at the photon energy at which the maximum amount of (negative) XMCD signal occurred.

Field-swept XDEPR spectra obtained at the copper and vanadium $L_3$ edges, respectively, are plotted in Fig. 2b, c. The percentage change Δ of the XMCD amplitude refers to the signal of the spectra at 3 T shown in Fig. 1a, c, scaled down linearly to the fields used in the XDEPR experiment. The quantity ΔXMCD follows the change of the net spin polarization ΔM upon microwave irradiation. A reduction in the sample magnetization, that is, a negative ΔM, results in a positive ΔXMCD since the XMCD contrast has a negative sign at the photon energy used for the detection. In the studied field range, a 1% change in the XMCD amplitude corresponds to a change of the transmitted X-ray intensity on the order of 10 ppm with a given circular polarization, which renders the experiments challenging.

As visible in Fig. 2b, the CuPc resonance with EPR transitions results in a characteristic XDEPR signature between 200 and 300 mT. While the two outer peak positions of the simulated and experimental spectra agree, the dominant peak of the experimental spectrum is not reproduced by the simulation. Since this experimental peak coincides with the resonance field of a free electron ($g_{free} = 2.002$), as indicated

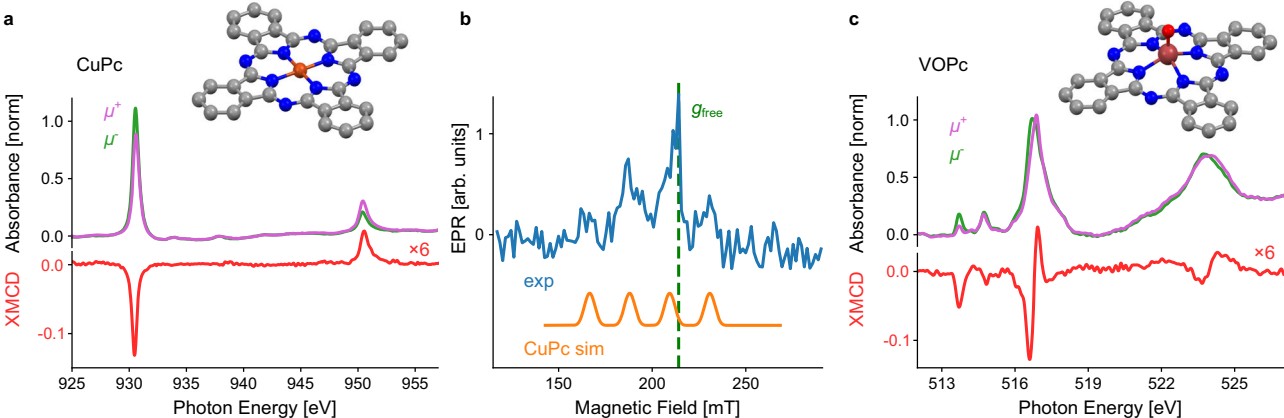

**Fig. 1 | Magnetic response of diluted copper(II)-phthalocyanine (CuPc) and vanadium(IV)-oxo-phthalocyanine (VOPc) films at low temperature (T ~ 5-6 K). a, c** Circularly polarized X-ray absorption spectra (XAS) and X-ray magnetic circular dichroism (XMCD) of the diluted films CuPc:H$_2$Pc (5%, 1 μm) and VOPc:TiOPc (10%, 0.5 μm), respectively, recorded at an applied magnetic field of 3 T in transmission mode. The molecular geometries of CuPc and VOPc are shown as ball-and-stick models. Color code: C – gray; N – blue; O – red; metal - brown. H atoms were omitted for clarity. **b** Field-swept pulsed EPR spectrum (blue) of CuPc:H$_2$Pc (5%, 1 μm) detected at $f = 6$ GHz using a relaxation filter (see Supplementary Note 7 in the SI) along with the simulated spectrum (orange) and the field position of g$_{free}$ (green dashed line).

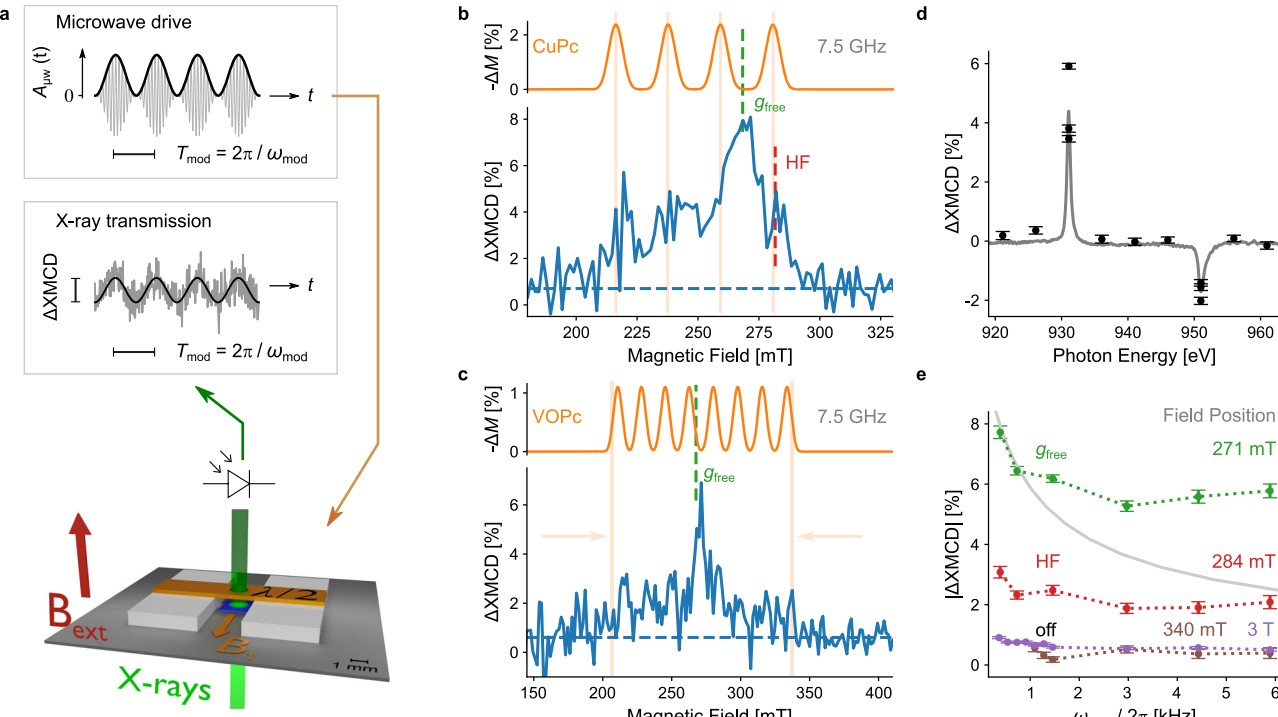

**Fig. 2 | Experimental setup and X-ray detected electron paramagnetic resonance (XDEPR). a** Scheme of the core experimental setup. The X-rays (green) pass through the sample (blue) inside the 7.5 GHz microwave resonator (orange), which is supported by spacers (white) on top of the baseplate (gray). The amplitude-modulated microwaves create a drop in the CuPc (VOPc) magnetization, which is detected as a change in the XMCD signal after demodulation using a lock-in amplifier. **b**, **c** Experimental XDEPR spectra with $f = 7.5$ GHz of the thin films CuPc:H$_2$Pc (5%, 1 μm) and VOPc:TiOPc (10%, 0.5 μm). The dashed blue lines indicate the finite, constant background, as explained in the text. Simulations of the relative change of magnetization $\Delta M$ due to the resonant microwaves with $B_1 = 0.1$ mT, $T_1 = 1$ μs, and $T_2 = 200$ ns are shown in orange. The vertical dashed lines indicate the free electron resonance field $g_{\text{free}}$, and the high-field (HF) shoulder in the case of CuPc. **d** Photon energy dependence of XDEPR at the free-electron resonance field recorded on sample CuPc:H$_2$Pc (20%, 0.4 μm) (black dots). The inverted XMCD spectrum measured at 3 T is superimposed and scaled to the XDEPR signal (gray solid line). **e** Dependence of XDEPR on the microwave amplitude modulation frequency recorded on CuPc:H$_2$Pc (5%, 1 μm) at four different magnetic fields, namely at the $g_{\text{free}}$ peak at 271 mT, at the high-field shoulder at 284 mT, and off-resonance at small (340 mT) and large (3 T) field values. The gray curve indicates the simulated behavior obtained with the literature-based parameters $T_1 = 0.1$ ms, $T_2 = 200$ ns, and $B_1 = 0.1$ mT. Error bars in (**d**, **e**) represent the noise level in demodulated time-domain data that were recorded for 6 min for each data point.

by the dashed green line, in what follows, we refer to this spectral feature as the '$g_{\text{free}}$ peak'. As opposed to the reference spectra at 3 T with (absolute) XMCD contrast of more than 10% with respect to the L$_3$ absorbance, the smaller magnetic fields required for resonance at 7.5 GHz result in an XMCD contrast at the few-percent level. Such small signals could be detected more efficiently by increasing the X-ray photon flux beyond 10 ph/ms/μm$^2$ as used for the XDEPR spectra. At a ten times larger photon flux, however, XDEPR spectra vanished within a few tens of minutes, even though the XMCD-detected magnetic moment of the Cu ion was preserved (see Supplementary Note 3 in the SI). We attribute this to a change in the magnetic environment of the Cu ion that is probed by XDEPR, as detailed further in the discussion.

For the XDEPR spectrum of the diluted VOPc thin film shown in Fig. 2c, the expected eight-line spectrum[33] (orange) is shown above the experimental data (blue). The dominant contribution to the XDEPR spectrum is also at $g_{\text{free}}$ (dashed green). An average time of 10 h was needed to obtain this spectrum, almost three times longer than for CuPc. Nevertheless, the experimental data indicate a broad spectral feature in the spectral window of the expected line shape, which is indicated by the orange arrows. Note that both the CuPc and the VOPc XDEPR spectra have a non-zero baseline (dashed blue) at a signal level of ~ 1% originating from a small modulation of the sample temperature caused by the microwave amplitude modulation (see Supplementary Note 4 in the SI).

The magnetic field-sweep spectra shown in Fig. 2b, c demonstrate the feasibility of X-ray detected EPR for the first time. To corroborate

this interpretation and the relation of the $g_{\text{free}}$ feature with the Cu and V ions, several control experiments have been performed. To further verify the probing of the element-specific magnetism in our XDEPR experiments, the photon energy dependence was measured at a constant magnetic field at the $g_{\text{free}}$ resonance. Figure 2d shows XDEPR data on the thin films Cu-3 [CuPc:H$_2$Pc (20%, 0.4 μm)] at different photon energies as well as the scaled, inverted XMCD spectrum at 3 T. As it is readily seen, the XDEPR signal follows the XMCD shape. Specifically, it exhibits the expected sign reversal at the L$_2$ edge compared to the L$_3$ edge. Besides the sign reversal with the X-ray polarization that is incorporated in the acquisition of $\Delta$XMCD, the sign reversal at the L$_{2,3}$ edges provides further evidence that the XDEPR signal stems from the reduction of the metal centers' longitudinal magnetic moments due to the resonant microwaves.

The $g_{\text{free}}$ resonance and the sensitivity to the photon flux constitute unexpected effects related to XDEPR that are treated further in the discussion section. Note that the presence of the $g_{\text{free}}$ resonance is not required for the XDEPR technique to work. Apart from these two aspects, the experimental $\Delta$XMCD contrast at the few-percent level raises the question of the spin relaxation times during XDEPR. In essence, the literature values of $T_1 = 0.1$ ms and $T_2 = 200$ ns for CuPc suggest a change of up to 15% in $\Delta$XMCD for the microwave field $B_1 = 0.1$ mT realized in our setup, which is almost an order of magnitude larger than in our experiments. The agreement between the experiment and simulations improves under the assumption of a shorter $T_1$, as exemplified by the simulated CuPc spectrum in Fig. 1b with $T_1 = 1$ μs. Accordingly, the following Section

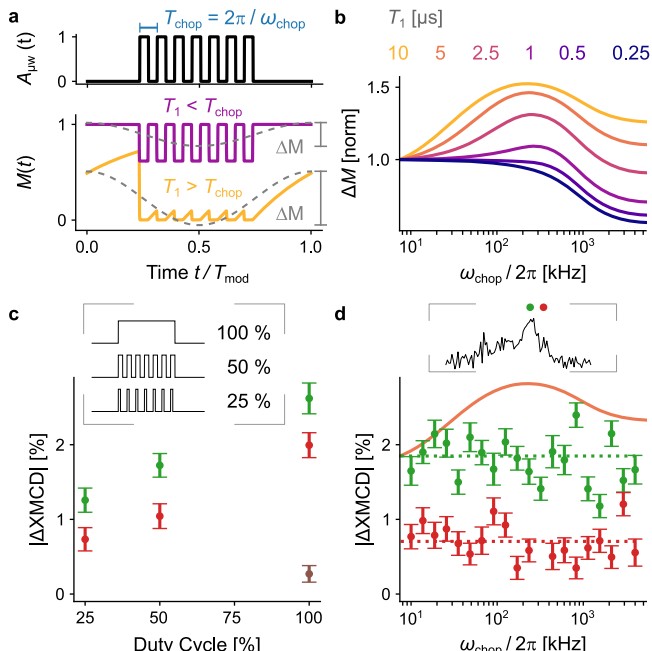

**Fig. 3 | Relaxation dynamics of electron spins. a** Modulation scheme for the chopping experiments as explained in the text. The microwave envelope $A_{\mu w}(t)$ (black) follows a two-frequency rectangular modulation with the fast chopping at $\omega_{chop}$ and the slow modulation at $T_{mod} = 2\pi/\omega_{mod}$. The simulated magnetization responses $M(t)$ are shown for two limiting cases of slow (magenta) and fast (light orange) $T_{chop}$ as compared to $T_1$. The dashed gray lines are the Fourier components at the lock-in demodulation frequency $\omega_{mod}$ with the $T_1$ dependent amplitude $\Delta M$. **b** Simulations at fixed $T_1$ showing the change in magnetization $\Delta M$ as a function of the chopping frequency $\omega_{chop}/2\pi$, using $T_2 = 200$ ns and $B_1 = 10$ μT. For each $T_1$ value, the data is normalized to $\Delta M$ at the smallest $\omega_{chop}$ value. **c** Experimental data showing the dependence of the XDEPR signal $\Delta$XMCD on the chopping duty cycle at constant chopping frequency $\omega_{chop}/2\pi = 10$ kHz for CuPc:H₂Pc (5%, 1 μm) at the central $g_{free}$ peak (green, 269 mT), at the high-field shoulder (red, 283 mT), and off-resonance (brown, 340 mT). The inset shows the envelopes $A_{\mu w}(t)$ for the different duty cycles. **d** Corresponding $\Delta$XMCD vs $\omega_{chop}/2\pi$ graph at the central $g_{free}$ peak (green) and at the high-field shoulder (red), as also indicated on the XDEPR spectrum in the inset. The dashed lines are the mean values of each data set. The solid orange curve is the simulated dependence from panel b for $T_1 = 5$ μs scaled to the experimental data at the $g_{free}$ peak. Error bars in (**c**, **d**) represent the noise level in demodulated time-domain data that were recorded for 6 min for each data point.

provides further experimental insight into the longitudinal relaxation dynamics.

## Determination of $T_1$

Accessing information about the spin dynamics is possible by varying the amplitude modulation frequency $\omega_{mod}$ of the microwaves[34,35]. Fig. 2e displays the effect of varying $\omega_{mod}$ for different applied magnetic fields, namely at the principal $g_{free}$ peak, at the high-field (HF) shoulder, and off resonance at 0.34 T and 3.0 T. At the lowest modulation frequency $\omega_{mod}/2\pi = 389$ Hz, the signal intensities at the different field values are in line with the XDEPR spectrum and its baseline in Fig. 2b. Upon increasing modulation frequency, only a small drop is visible, and the signal becomes frequency independent beyond $\omega_{mod}/2\pi = 1$ kHz. The initial decrease is ascribed to the instrumental response of the X-ray photodetector (see Supplementary Note 5 in the SI). In general, the dependence on the modulation frequency can be rationalized as follows: When the longitudinal spin relaxation time $T_1$ is shorter than the modulation period ($T_1 \ll 1/\omega_{mod}$), the spin polarization can follow the modulation. In the opposite limit of a modulation period that is short compared to $T_1$ ($T_1 \gg 1/\omega_{mod}$), the excited spins remain

in saturation without noticeable modulation of the polarization, such that the XDEPR signal vanishes (see Supplementary Note 5 in the SI). The results in Fig. 2e, therefore, suggest that the spins can follow the drive for all applied modulation periods, even down to the shortest period of $T_{mod} = 166$ μs. This timescale is comparable to the expected $T_1$, which is on the order of 0.1 ms based on literature[8] consistent with our pulsed EPR experiments of the same sample at 5 K (see Supplementary Note 7 in the SI). However, the gray line shows the variation with the modulation frequency that one would expect for $T_1 = 0.1$ ms showing a clear drop, which is not compatible with the experimental results.

To access faster $T_1$ relaxation dynamics, an adapted experiment was devised as described in Fig. 3a using a two-frequency amplitude modulation scheme of the driving microwave envelope $A_{\mu w}(t)$. This consists of (i) a slow rectangular modulation at $\omega_{mod}$ for the lock-in detection of the $\Delta$XMCD signal and (ii) a fast rectangular modulation at $\omega_{chop}$, which is referred to as the chopping frequency. The expected behavior can be rationalized as follows: In the limit where the longitudinal relaxation time is short with respect to the chopping period $T_1 \ll T_{chop} = 2\pi/\omega_{chop}$, the spin polarization follows the fast-chopping envelope (magenta), whereas in the other extreme $T_1 \gg T_{chop}$, the slowly relaxing spins remain saturated, as shown as a light orange solid line. These induced dynamics influence the amplitude $\Delta M$ of the Cu magnetization at the slow modulation frequency $\omega_{mod}$, which is then retrieved experimentally by demodulation at $\omega_{mod}$ by the lock-in amplifier (dashed gray lines): Here, $\Delta M$ depends on $T_1$ and $T_{chop}$ as illustrated in Fig. 3a, hence from these experiments $T_1$ can be inferred using suitable modeling. Figure 3b shows the simulated dependence of $\Delta M$ as a function of $\omega_{chop}$ for different fixed $T_1$ times. The curves are normalized to the initial values at the smallest $\omega_{chop}$. The $T_1$-dependent initial rise with $\omega_{chop}$ is due to the onset of saturation. At the fastest $\omega_{chop}$, a reduction of $\Delta M$ is observed due to the influence of $T_2$ (see Supplementary Note 6 in the SI).

Experimental results for the diluted CuPc film Cu-1 are shown in Fig. 3c, d, where $\Delta$XMCD recordings at the $g_{free}$ peak (green) and at the high-field shoulder (red) are shown. To outline the method, Fig. 3c shows the influence of the duty cycle of the chopping envelope on $\Delta$XMCD at constant $T_{chop} = 100$ μs. In the absence of chopping, at 100% duty cycle (see inset), $\Delta$XMCD data at the two spectral positions and at the extra data point off-resonance (brown) are smaller than in Fig. 2b, e due to the purposely reduced microwave power. The introduction of chopping at 50% or 25% duty cycle reduces $\Delta$XMCD progressively, which suggests that the spin polarization follows the chopped envelope, hence $T_1 \ll T_{chop}$. For the examination of shorter chopping intervals, Fig. 3d shows the experimental dependence of $\Delta$XMCD on the chopping frequency $\omega_{chop}$ at 50% duty cycle, together with the simulated curve for $T_1 = 5$ μs from Fig. 3b (orange) that was scaled to the data at the $g_{free}$ peak. The experimental data do not show a pronounced change with $\omega_{chop}$, which suggests a relaxation time in the lower microsecond range $T_1 < 5$ μs. Note that this is a regime where it becomes difficult to precisely extract $T_1$ from the presented chopping data, since the changes in $\Delta$XMCD become small and, in addition, depend critically on both $T_2$ and $B_1$ (see Supplementary Note 6 in the SI).

In summary, the observation of longitudinal relaxation times that are significantly faster than the literature value of 0.1 ms by two complementary XDEPR approaches provides evidence for enhanced $T_1$ relaxation pathways. Note that temperature effects on $T_1$, as reported for CuPc[8,36], were minimized by purposely reducing the microwave power in the chopping experiments. Furthermore, the high frequency chopping at a 50 %-duty-cycle leads to an additional power reduction. Therefore, it is very unlikely that the acceleration of $T_1$ can be explained exclusively through a temperature increase induced by microwave irradiation.

## Discussion

The presented XDEPR experiments are challenging mainly due to the low signal levels which are significantly weaker than what is obtained in XFMR. The reason behind this is that the narrow spectral holes that are created upon microwave irradiation only result in a very small change of the longitudinal magnetic moment of the spin ensemble detected by XMCD. In the following several unexpected and noteworthy observations that were presented in the previous Section will be discussed in more detail: (i) The XDEPR signal decreases with increased X-ray photon flux. (ii) A strong XDEPR response appears at $g_{free}$ and (iii) the longitudinal $T_1$ relaxation times are shorter than without X-ray illumination.

We emphasize that upon X-ray illumination, no changes in the static XMCD spectra at different fields are observed, which excludes photoreduction and beam damage effects. Hence, observations (i-iii) in XDEPR imply the presence of X-ray-induced changes invisible in the conventional X-ray absorption spectra, that is, not affecting the magnetic moment of the metal center's spin. They occur exclusively in XDEPR and indirectly probe the magnetic environment of the investigated metal spin centers.

To understand our observations, the trapping of secondary electrons at defect sites and the formation of radicals in the vicinity of the metal spins need to be considered. Secondary electrons are generated by the decay of the X-ray excited core-hole state with energies of a few $eV$[37]. For the studied phthalocyanine complexes, it is known that the conjugated $\pi$ orbitals of the phthalocyanine ligand can host radicals, that is, unpaired electrons, giving rise to an EPR peak close to $g_{free}$[38–41]. Studies involving photo-generated radicals in $H_2Pc$[39,40] suggest the presence of a variety of trapping sites able to trap electrons or holes. Either localized and long-lived (deep traps) or short-lived radical states delocalized over several Pcs (shallow traps) can be formed. The concentration of the trapped spins and produced radicals depends on the incident X-ray photon flux.

### Sensitivity to the photon flux

For the investigated paramagnetic CuPc and VOPc complexes, the presence of such radical states does not alter the oxidation state of the metal center, as evidenced by magnetic susceptibility measurements[42–44]. This is also confirmed by our Cu L-edge spectra in which a photoreduction effect would give rise to the appearance of a second peak shifted toward higher photon energy[45]. Despite the absence of photoreduction, the EPR signal of the metal center surrounded by a radical localized on its Pc ligand was shown to be reduced or even completely lost in numerous studies[44,46–48]. This is analogous to our observation of CuPc at high photon flux, where the Cu magnetic moment was preserved, while the EPR spectrum was no longer detectable (see Supplementary Note 4 in the SI). We ascribe this phenomenon to a high concentration of radicals generated by the incoming X-rays. In contrast, at a low photon flux, the concentration of X-ray excited radicals is small enough to allow for the observation of XDEPR.

### Appearance of a strong XDEPR response at $g_{free}$

Independently, there can be short-lived radical states delocalized over a few Pc ligands, similar to the photo-excited radicals observed with visible light in $H_2Pc$ with lifetimes on the order of 10 $\mu s$[49,50]. We consider these delocalized radicals to be responsible for the appearance of the $g_{free}$ peak in the XDEPR spectra. In essence, the coupled radical-metal spin system does not have strictly separable spin states. In the spectral regions where the radical and metal spectra overlap, dipolar or exchange spin-spin coupling mixes the spin states by off-diagonal terms in the spin Hamiltonian, such that the radical and metal spin states are no longer separable. As a result, the XMCD-detected longitudinal magnetic moment no longer exclusively represents the spin polarization of the metal center's spin transitions. This introduces a pathway for the $g_{free}$ peak of the radical to be observed in the XMCD signal that is specific to the metal ion. Note that without the presence of photo-excited radicals, the $g_{free}$ peak would vanish, but the XDEPR signal measured at the Cu $L_3$ edge would still remain. A detailed theoretical analysis of this spectral leakage pathway is provided in Supplemental Note 9 in the SI.

### Reduced longitudinal $T_1$ relaxation times

To understand the $T_1$, which is shortened in our XDEPR measurements compared to previously published EPR results, we need to consider that dynamic processes can occur which cause thermalization of the spin polarization. For surface-cast molecular and single-atom magnets, relaxation channels with a timescale of several seconds related to the direct scattering between secondary electrons and the probed spins have been observed[21,51]. In our XDEPR experiments, we consider the photon flux of 10 ph/ms/$\mu m^2$ to be too small to cause the observed microsecond relaxation times via such a direct spin-secondary electron interaction that is spatially localized around the X-ray excited core-hole. Instead, we attribute a pivotal role to the trapping sites in our samples that facilitate less direct relaxation channels between the spins and trapped secondary electrons. Potential relaxation channels could be spin-lattice relaxation by phonons excited by hot electrons, mediated by the electron-phonon coupling[52], or relaxation via charge fluctuations that induce local magnetic fields at the Larmor frequency[53,54]. In analogy to the $g_{free}$ peak, delocalized radical states could be a probable source of fluctuating charges that modulate the radical-metal spin-spin coupling on picosecond timescales. In this scenario, the delocalization and lifetime of these states can promote additional relaxation pathways to the ensemble of metal centers. Accordingly, the X-ray generated radicals can be viewed as a metastable network of fluctuating spins that gives rise to the $g_{free}$ peak and whose picosecond dynamics relax the localized metal spins (see also Supplementary Note 9 in the SI).

In conclusion, we have demonstrated for the first time the feasibility of X-ray-detected EPR in the soft X-ray range on dilute thin films of molecular quantum bit candidates. This opens up novel avenues toward element-specific magnetic measurements with ultrahigh spectroscopic resolution. We have identified several X-ray-specific effects, which suggest the presence of X-ray-generated radicals and the filling of charge traps by secondary electrons. These effects need to be considered in future experimental efforts in this research direction since they critically influence the resulting sensitivity of XDEPR. As an example, we would expect fewer charge-trapping sites in inorganic solid-state substrates than in the phthalocyanine thin films investigated in this study. The overall X-ray dose could also be reduced in pulsed approaches that separate microwave excitation and X-ray detection in the time domain, which requires an X-ray timing structure as provided by X-ray choppers[55] or X-ray free-electron laser facilities[56]. Ultimately, such pulsed approaches would allow accessing the quantum-coherent properties of the probed spins. Furthermore, using the highly sensitive electron-yield detection mode, the XDEPR investigation of monolayer-thin magnetic materials could be possible.

## Methods

### Thin film sample growth

The molecular films were grown on 800 nm or 1500 nm thin aluminum foils (Goodfellow Cambridge Ltd, AL000210 or AL000271) in a dedicated deposition chamber with a base pressure of $p_{base} \sim 10^{-8}$ mbar. The targeted dilution of paramagnetic and diamagnetic phthalocyanine materials, respectively, was achieved by co-evaporation using a three-cell organic evaporator (Kentax, TCE-CS 5x) and a quartz-crystal microbalance to calibrate the evaporation rate. In order to achieve net

growth rates of ~ 2 nm/min, alumina crucibles (ALB materials, TAC-012) were used for the diamagnetic hosts $H_2Pc$ and TiOPc, while standard quartz crucibles were used for CuPc and VOPc. A 10 nm-thick templating layer of 3,4,9,10-perylenetetracarboxylic dianhydride (PTCDA)[57] was deposited onto the aluminum foil prior to the deposition of the Pc materials. All source chemicals were obtained from Sigma Aldrich, except for TiOPc, which was acquired from TCI chemicals. The chemicals were used as received and degassed inside the evaporator prior to film deposition.

The samples prepared for the XDEPR experiments are listed in Supplementary Table 2. The quality and orientation of the deposited films were verified by X-ray linear dichroism (XLD) at the nitrogen K-edge, as detailed in a preceding study[27]. In fact, the VOPc thin film used here was prepared alongside the diluted 500 nm VOPc film studied in ref. 27.

## Microwave setup and resonator

The microwave drive was provided by an open-source EPR spectrometer[58], which has been frequency-translated to the required resonance frequency at 7.5 GHz. A detailed description of this setup is provided in Supplementary Note 1 in the SI. The resonator ($f \sim 7.5$ GHz) is formed by a sheet of aluminum suspended above a conducting *ground plane* by 1 mm thick Teflon spacers. This half-wave microstrip resonator[59] supports a standing wave with maximum electric fields at its non-contacted edges and maximum magnetic fields around the center of the strip. The latter resonantly excites the sample supported on an ultrathin aluminum foil, which is placed onto the ground plane for optimized thermal contact with the cold finger of the cryostat. The resonator is driven by a feed line attached to an XYZ manipulator, by which the contactless coupling can be optimized in situ (see also Supplementary Note 1 in the SI for more details).

The X-rays were transmitted through aligned holes in the resonator (see Fig. 2a) and in the ground plane. The hole in the ground plane was covered by the sample grown on X-ray transparent sheets of 800 nm or 1500 nm thin aluminum foils. This thickness provides sufficient microwave conductivity at cryogenic temperatures, such that microwave field perturbations around the X-ray transmission hole are avoided.

While the X-Treme cryostat coldfinger reaches temperatures below 3 K for ordinary XMCD studies, the sample temperature in the present work is elevated to ~ 6 K in the microwave resonator (see Supplementary Note 3 in the SI). The EPR performance of the contactless coupling employed here has been benchmarked with both pulsed and continuous-wave EPR spectroscopy using reference compounds inside the X-Treme endstation and with a dedicated test setup (see also Supplementary Note 8 in the SI). These experiments yielded a power-to-field conversion factor[60] $\Lambda$ ~ 50 μT/ W.

## X-Ray Spectroscopy

All X-ray experiments were performed at the EPFL/PSI X-Treme beamline[61] at the Swiss Light Source, situated at the Paul Scherrer Institut in Switzerland. The lateral size of the incoming beam was 0.5 mm × 0.6 mm, and the photon flux was adjusted to 10 ph/ms/μm² unless explicitly indicated otherwise. The intensity of the X-rays transmitted through the sample and resonator was measured by a photodiode and normalized to the incoming beam intensity. The applied magnetic field was always collinear with the X-ray beam and perpendicular to the sample and the resonator plane.

**XMCD spectra.** XMCD spectra were obtained from normalized and background-subtracted absorbance spectra as $(y_c^+ - y_c^-)/2$. The raw absorbance spectra were computed by $y_a^{\pm} = -\log(y^{\pm}/a^{\pm})$, where $a^{\pm}$ is the pre-edge intensity of the raw experimental data $y^{\pm}$. Subtraction of a linear baseline and subsequent normalization to the $L_3$ edge amplitude yielded $y_c^{\pm}$ from $y_a^{\pm}$. Normalization and baseline correction were done independently of the X-ray polarization and deduced from the unpolarized XAS $(y_a^+ + y_a^-)/2$.

**XMCD magnetization curves.** XMCD magnetization curves as a function of the external longitudinal magnetic field were recorded between 0 T and 6.8 T, with separate $\mu^-$ and $\mu^+$ circular polarizations in the forward sweep and the backward sweep, respectively. During the field ramp, the photon energy was continuously toggled between $e_1$ and $e_2$ to probe either XMCD at the $L_3$ edge or the pre-edge absorption, respectively. Raw absorbance values $y_a^{\pm}$ were computed from raw data $y^{\pm}$ via $y_a^{\pm} = -\log(y^{\pm})$. XMCD was extracted upon pre-edge normalization, e.g., $(y_a^+(e_1)/y_a^+(e_2) - y_a^-(e_1)/y_a^-(e_2))$.

**Field-swept XDEPR spectra.** Field-swept XDEPR spectra as a function of the applied magnetic field were recorded at a fixed photon energy tuned to the maximum XMCD contrast. The AC variation of the transmitted light due to the amplitude modulation of the microwaves at $\omega_{mod}/2\pi = 389$ Hz was demodulated with a lock-in amplifier. The demodulation bandwidth was 1 Hz, and the demodulated in-phase and quadrature components were relayed to auxiliary analog recording channels of the X-Treme beamline.

A single field scan took 10 mins for CuPc and 20 mins for VOPc. Several field scans were accumulated and averaged. In order to extract the XDEPR signal, the X-ray polarization was cycled from scan to scan between $\mu^+$ and $\mu^-$. Furthermore, the microwave modulation phase $\varphi_{mod}$ was cycled by incrementing it in steps of $\pi/2$ with respect to a fixed demodulation oscillator. This results in a cyclic permutation of the in-phase and quadrature components detected by the lock-in amplifier. This phase cycle suppresses any spurious contributions originating from modulation pathways that are not caused by amplitude modulation of the microwaves.

The XDEPR data averaged in this way had units of voltages. In order to reflect the relative change of the longitudinal magnetic moment, the amplitude of the XMCD modulation was expressed as percentage change $\Delta$XMCD with respect to the field-dependent static XMCD amplitude. The static XMCD amplitude for normalization was extracted from XMCD experiments at 3 T and linearly scaled to the appropriate field value.

**Constant-field XDEPR data.** Constant-field XDEPR data were obtained using the same experimental protocol as XDEPR spectra, namely with a combined cycle of the X-ray polarization and of the modulation phase. Each $\Delta$XMCD data point was acquired for a total time of 6 min and was computed from time-domain recordings of demodulated lock-in data, which had a demodulation bandwidth of 0.5 Hz in constant-field experiments. Each constant-field XDEPR data point includes an error bar, which is the standard deviation of the demodulated time-domain recording from its mean value.

## EPR simulations

Spectral lineshapes of CuPc and VOPc were calculated using the EasySpin[62] software package employing literature parameters for the spin Hamiltonians of CuPc[32] and VOPc[33]. For CuPc $g$-factors of $g_{\parallel} = 2.157$, $g_{\perp} = 2.039$, and the hyperfine interaction with the $^{63}Cu$ nuclear spin of $A_{\parallel} = -648$ MHz and $A_{\perp} = -83$ MHz were used. The couplings to surrounding $^{14}N$ nuclear spins were included in the calculations but not resolved in the resulting spectra, due to a convolution broadening of 2.5 mT required to reproduce the line shape of similar reported CuPc thin films[8]. For VOPc, we used $g$-factors of $g_{\parallel} = 1.988$, $g_{\perp} = 1.987$, and $A_{\parallel} = 482$ MHz, $A_{\perp} = 171$ MHz for the hyperfine interaction with the $^{51}V$ nuclear spin. Surrounding $^{14}N$ nuclear spins were included, but unresolved due to convolution broadening of

2.5 mT. For both CuPc and VOPc, the spectral line shapes were simulated for the magnetic field aligned parallel to the interaction tensors.

XDEPR spectra were simulated by convolution of the field-domain EPR lineshape described above with the analytical steady-state 'hole pattern' in the longitudinal magnetization component that results in a driven two-level system with the longitudinal and transverse relaxation times $T_1 = 1\,\mu s$ and $T_2 = 200\,ns$, respectively, and with $B_1 = 100\,\mu T$. For dynamical quantities, namely the dependence of the spin polarization on the modulation frequencies $\omega_{mod}$ and $\omega_{chop}$, the Bloch equations were integrated numerically in the time domain using the solve_ivp package in Python[63,64]. In order to compute steady-state dynamics throughout one modulation period $T_{mod}$, the integral over $T_{mod}$ has been calculated over multiple iterations. Each iteration started with the final state of the previous iteration until the magnetization trajectory converged to a final steady-state solution. As detailed further in Supplementary Note 5 in the SI, this simulation approach reproduces analytical descriptions in the limit of weak driving fields in the absence of chopping, which are the conditions where analytical solutions have been elaborated[34]. The extension of these simulations of the Bloch equations toward coupled spins has been performed using the SPIn DYnamics ANalysis (SPIDYAN) package[65] as described in Supplementary Note 9 in the SI.

### EPR spectroscopy of CuPc

The EPR spectrum of CuPc thin-film Cu-1 has been obtained by pulsed EPR spectroscopy at 6 GHz on a dedicated setup described in Supplementary Note 7 in the SI. The thin-film sample was exactly the same as used for XDEPR and the EPR experiments at 6 GHz were performed after completion of the X-ray experiments. Additionally, for calibration of pulse flip angles, a small flake of the Koelsch radical $\alpha,\gamma$-bisdiphenylen-$\beta$-phenylallyl (BDPA) was placed on top of the thin-film sample, which has a very narrow linewidth of ~ 0.1 mT at $g_{free}$. A relaxation filter was used to reduce the contributions of the probe background and the BDPA reference to the EPR spectrum (see Supplementary Note 7 in the SI).

## Data availability

The data generated in this study have been deposited in the Zenodo database under https://doi.org/10.5281/zenodo.14204505.

## Code availability

The code for data evaluation and for simulations developed in this study have been deposited in the Zenodo database under https://doi.org/10.5281/zenodo.14204505.

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

## Acknowledgements

A.D. and J.D. thank the Swiss National Science Foundation (grant no. 200020_182599) and the Paul Scherrer Institut (CROSS project "Toward Quantum Technologies with X-rays and Muons") for financial support. The authors thank Simone Finizio and Jörg Raabe for fruitful discussion and Stefan Zeugin for technical support. Daniel Klose and Gunnar Jeschke are acknowledged for a temporary loan of a cryostat used in our pulsed EPR setup.

## Author contributions

A.D., G.B., S.R., H.B., Z.S. and J.D. conceived and planned the experiments. Z.X., V.R. and A.D. prepared the samples, and A.D. and J.D. performed the experiments. A.D. analyzed and visualized the data. Z.S. and J.D. acquired funding. A.D. and J.D. wrote the manuscript with input from all authors.

## Competing interests

The authors declare no competing interests.
