## [Transparent Peer Review file · Nature Communications]

Element-Specific X-Ray Detection of Electron Paramagnetic Resonance in Thin Films of Quantum Bits

Corresponding Author: Dr Andrin Doll

Version 0:

Reviewer comments:

Reviewer #1

(Remarks to the Author)

The title of this manuscript has really fascinated me. The development of spatially averaged but element-selective and, in principle, surface-sensitive detection of EPR magnetic resonance is of great interest for the development of quantum technologies based on spin. The topic is certainly relevant for a broad audience journal.

I have to admit that I have been somehow disappointed by the results even if I fully acknowledge the value of this initial effort.

1. The method. The authors use variable amplitude of the microwave. They quote ref. 31 that has nothing to do with this method that actually is better described in EPR book such as:

Computational and Instrumental Methods in EPR by Christopher J. Bender Lawrence J. Berliner. This is to inform the reader that it is a well-established, though not often employed, method.

2. The addition here is to use XMCD to detect the change in longitudinal magnetization which for a spin $S=1/2$ corresponds to the absorption of the microwave.

3. The sensitivity of the XMCD detection appears rather low, compared to the standard sensitivity of EPR. Indeed, a monolayer of radical spins can be detected in standard EPR, and for sure, 'thick' samples of the molecules investigated here are better investigated with traditional EPR (see ref. 8). As a result, the VOPc spectrum is hardly detectable and also the CuPc one is very noisy.

4. The authors work with relatively concentrated species and their result should be compared with T1 extracted from AC susceptometry rather than EPR. Even for concentration of 20% CuPc has very long T1, see 10.1021/acs.inorgchem.0c02573. T1 seems to be poorly affected by concentration of spins, thus it is hard to understand why it should collapse as an effect of the formation of radicals. Even in coupled spin systems, T1 remains rather long (10.1021/jacs.1c02417).

Thus their justification remains quite speculative.

5. Element selectivity. Here is the real point of strength of the proposed methodology. Unfortunately the authors do not provide an explanation and an estimate of how the XMCD intensity at the Cu edge can be influenced by the mechanism of a weak coupling with an organic radical formed by the secondary e- following the X-ray absorption.

If the resonance occurs at $g=2$, it means that the contribution of the 3d spin is very limited and the exchange interaction is rather weak. Otherwise, one would see a spin with a g value that is a linear combination of the individual spins. How such a weak exchange would influence the XMCD intensity? Here, the authors should provide at least a semiquantitative estimation. In addition, we are again in the situation of 10.1021/jacs.1c02417, which does not experience a collapse of T1.

In conclusion, I encourage the authors to work along this interesting line. However, the manuscript seems too premature for publication in a high-impact journal. On the one hand, the signals obtained are not at all remarkable compared to traditional EPR. On the other hand, the authors do not justify the observed signal at $g=2$ and do not propose a sound model for the collapse of T1.

A significant revision and re-evaluation appear necessary.

Reviewer #2

(Remarks to the Author)

The manuscript entitled „Element-Specific X-ray Detection of Electron Paramagnetic Resonance in Thin Films of Quantum

Bits“ by A. Doll et al. deals with the detection of coherent spin superposition states by employing X-ray detected EPR in combination with X-ray Magnetic Circular Dichroism (XMCD).

For classical ferromagnetic spin systems an analogous approach combining XMCD and X-ray Ferromagnetic Resonance is an established approach to study the spin dynamics element specific.

The challenge to study (diluted) quantum coherent paramagnetic states is that the signal compared to XFMR is very low.

The authors managed to apply an improved detection scheme already used for XFMR to overcome this problem in a sufficient way to demonstrate the possibility to probe XDEPR. As model systems diluted CuPc and VPc molecules, grown on thin aluminium films comprising Cu²⁺ and V⁴⁺ ions have been used.

Element specific XDEPR is demonstrated for both systems, using the energy showing maximum XMCD as x-ray excitation energy. A rather strong XDEPR response is found at g_{free} , and reduced longitudinal T₁ relaxation times as compared to the conventional EPS measurements, which might be explained due to thermalization of the spin polarization.

Whereas there is almost no new information on the materials used, this is the first successful demonstration of element specific x-ray detected EPR. This approach will be very valuable for experiments on a number of diluted paramagnetic centers of single molecular magnets or molecular qubits for instance. Such experiments will help to better understand the underlying quantum-coherent states of these highly interesting quantum systems.

Therefore, I recommend publication of this manuscript in Nature Communications as first experimental realization of element specific XDEPR.

Version 1:

Reviewer comments:

Reviewer #1

(Remarks to the Author)

I thank the authors for having taken into consideration my comments and improved the manuscript.

I am in favor of its publication overall. However, it is clear from the figure 2b and 2c that only having the computed EPR spectrum it is possible to interpret the spectra. In these conditions, the technique has limited added value to investigate thin films of molecular spins. The authors should consider adding a clear statement about the need to increase the technique's sensitivity and possibly propose a roadmap for it.

I have read the explanation they provide about the appearance of a $g=2$ feature in the Cu-edge XMCD detected EPR. The model sounds fine, but if I am not wrong, the simulation in SI refers to the EPR signal. I have not seen the equivalent simulation in the change of magnetization of the Cu²⁺ spin, which is actually what is measured.

I might be wrong here, but I am afraid that most readers will miss this point.

Point-by-point response to the comments of the Reviewers. Color code: Reviewers: black normal; authors: blue italic.

Reviewer #1:

The title of this manuscript has really fascinated me. The development of spatially averaged but element-selective and, in principle, surface-sensitive detection of EPR magnetic resonance is of great interest for the development of quantum technologies based on spin. The topic is certainly relevant for a broad audience journal.

I have to admit that I have been somehow disappointed by the results even if I fully acknowledge the value of this initial effort.

1. The method. The authors use variable amplitude of the microwave. They quote ref. 31 that has nothing to do with this method that actually is better described in EPR book such as: Computational and Instrumental Methods in EPR by Christopher J. Bender Lawrence J. Berliner. This is to inform the reader that it is a well-established, though not often employed, method.

We thank the Reviewer for pointing out this error in referencing. It is true that the amplitude modulation is not directly related to the XDEPR technique, and that it has been performed before, while XDEPR is entirely new. We have revised the manuscript accordingly and added the suggested book chapter written by Sushil Misra and also a publication by Vadim Atsarkin and co-workers (references 34 and 35 in the revised manuscript).

2. The addition here is to use XMCD to detect the change in longitudinal magnetization which for a spin $S=1/2$ corresponds to the absorption of the microwave.

Indeed, XMCD detects the change in longitudinal magnetization (measured along the X-ray beam direction) upon resonant microwave absorption.

3. The sensitivity of the XMCD detection appears rather low, compared to the standard sensitivity of EPR. Indeed, a monolayer of radical spins can be detected in standard EPR, and for sure, 'thick' samples of the molecules investigated here are better investigated with traditional EPR (see ref. 8). As a result, the VOPc spectrum is hardly detectable and also the CuPc one is very noisy.

The Reviewer is right that a monolayer of radical spins can be detected in standard EPR. However, this is not true for surface-adsorbed transition-metal based spin- $\frac{1}{2}$ systems, which are investigated in our present work because of their broader EPR line widths. Furthermore, our technique provides element specificity, which would allow distinguishing different species of qubits. This would be difficult or impossible to realize using standard EPR.

In the work described in Ref. 8 a standard EPR spectrometer was used with a resonator incorporating multiple sliced films. Consequently, the effective area of these films amounts to several cm^2 . In our study, the probed surface of XDEPR is on the order of only 1 mm^2 , thus more than two orders of magnitude lower than in ref 8 with standard EPR. Also note that our samples are grown on conductive Al foils for thermalization under vacuum. It would not be possible to stack such films on top of each other for the investigation in a standard EPR spectrometer due to the skin effect. Accordingly, in our study, we have performed pulsed EPR with a mm-sized resonator that is dedicated for our thin films on conductive substrates.

In addition we emphasize that the present work is a proof of principle of this technique, currently working in X-ray transmission mode. Once the much more surface sensitive total electron yield (TEY) mode can be employed, which is routinely used in standard (non-EPR) X-ray absorption spectroscopy, extremely enhanced sensitivities can be expected. As mentioned in the paper, this still requires solving a number of technical issues but does not seem to be completely out of reach.

4. The authors work with relatively concentrated species and their result should be compared with T_1 extracted from AC susceptometry rather than EPR. Even for concentration of 20% CuPc has very long T_1 , see 10.1021/acs.inorgchem.0c02573. T_1 seems to be poorly affected by concentration of spins, thus it is hard to understand why it should collapse as an effect of the formation of radicals. Even in coupled spin systems, T_1 remains rather long (10.1021/jacs.1c02417). Thus their justification remains quite speculative.

We thank the Reviewer for pointing this out and we now also cite AC susceptometry data for T_1 relaxation. Based on the quoted literature results one would not expect the T_1 time to be shortened significantly in our experiments. However, as a matter of fact, our XDEPR results are inconsistent with such long T_1 times, indicating that there are further mechanisms that contribute to the reduction of T_1 in our experiments, as compared to the ones reported in the literature.

Note that the coupled spin systems studies in (10.1021/jacs.1c02417) include a pair of confined spins that is held at a well-defined distance via molecular spacers with saturated bonds to avoid exchange coupling. By contrast, we are dealing with coupled spin systems consisting of the slowly relaxing Cu spin and fast relaxing electron spins diffusing across the semiconducting π orbitals of the Pc rings. In this regard, on page 8071 of the very JACS paper 10.1021/jacs.1c02417 it is stated that "... cases where there is a fast relaxing qubit and a slow relaxing qubit, the spin flips of the fast relaxing one are a major source of magnetic noise."

In the revised manuscript, we have added a new Section in the Supporting Information about the theoretical foundations of the g_{free} peak, where we elaborate further on the type of spin system that is formed under X-ray irradiation (c.f. the new Section "VIII. EXTENDED THEORY ON G-FREE PEAK"). We hope that this clarifies our viewpoint and allows for a better distinction between the metastable spin pairs that are formed in our samples and the numerous examples of long- T_1 spin pairs that are frequently studied by EPR using double electron-electron resonance (DEER) techniques.

5. Element selectivity. Here is the real point of strength of the proposed methodology. Unfortunately the authors do not provide an explanation and an estimate of how the XMCD intensity at the Cu edge can be influenced by the mechanism of a weak coupling with an organic radical formed by the secondary e^- following the X-ray absorption. If the resonance occurs at $g=2$, it means that the contribution of the 3d spin is very limited and the exchange interaction is rather weak. Otherwise, one would see a spin with a g value that is a linear combination of the individual spins. How such a weak exchange would influence the XMCD intensity? Here, the authors should provide at least a semiquantitative estimation. In addition, we are again in the situation of 10.1021/jacs.1c02417, which does not experience a collapse of T_1 .

We are happy that the Reviewer states "Here is the real point of strength of the proposed methodology." As suggested by the Reviewer, we have performed quantum-mechanical simulations of coupled two-spin systems (radical spin and Cu metal spin) with the detection of only one of the two spins. The simulations indeed exhibit the expected signature of the radical spin in the element specific signal when only the metal spin is detected. The underlying reason is that the spin states are mixed when the coupling becomes significant with respect to the frequency separation between the spins. The resultant non-separability of the coupled spin states directly manifests as a contribution at g_{free} .

We hope that the newly added Section in the Supporting Information clarifies the origin of the g_{free} peak. We do admit that it is difficult to obtain a fully quantitative understanding of the g_{free} peak because of the delocalized and dynamic nature of the metastable network of trapped electrons that surround the metal ions probed by the X-rays. As we outlined in the added Section, the principal challenges for a fully quantitative understanding are (i) the strength of the spin-spin coupling for delocalized states and (ii) the motional averaging effects due to the involved dynamics and the T_1 relaxation of the coupled spin.

In conclusion, I encourage the authors to work along this interesting line. However, the manuscript seems too premature for publication in a high-impact journal. On the one hand, the signals obtained are not at all remarkable compared to traditional EPR. On the other hand, the authors do not justify the observed signal at $g=2$ and do not propose a sound model for the collapse of T_1 . A significant revision and re-evaluation appear necessary.

We hope that our point-by-point responses above as well as the added theory convince the Reviewer that our work is not too premature for publication. We again want to point out that all the points that the Reviewer mentions are related to delocalized trapped secondary electrons created by the X-rays, namely the presence of the g_{free} peak, the reduction of T_1 and the associated loss of sensitivity by an order of magnitude since faster relaxing spins are more difficult to saturate. Future research will focus on how to reduce the concentration of trapped electrons by both changes in sample environment and the experimental protocol.

Reviewer #2 (Remarks to the Author):

The manuscript entitled „Element-Specific X-ray Detection of Electron Paramagnetic Resonance in Thin Films of Quantum Bits“ by A. Doll et al. deals with the detection of coherent spin superposition states by employing X-ray detected EPR in combination with X-ray Magnetic Circular Dichroism (XMCD). For classical ferromagnetic spin systems an analogous approach combining XMCD and X-ray Ferromagnetic Resonance is an established approach to study the spin dynamics element specific.

The challenge to study (diluted) quantum coherent paramagnetic states is that the signal compared to XFMR is very low. The authors managed to apply an improved detection scheme already used for XFMR to overcome this problem in a sufficient way to demonstrate the possibility to probe XDEPR. As model systems diluted CuPc and VPc molecules, grown on thin aluminium films comprising Cu^{2+} and V^{4+} ions have been used.

Element specific XDEPR is demonstrated for both systems, using the energy showing maximum XMCD as x-ray excitation energy. A rather strong XDEPR response is found at g_{free} , and reduced longitudinal T_1 relaxation times as compared to the conventional EPS measurements, which might be explained due to thermalization of the spin polarization.

Whereas there is almost no new information on the materials used, this is the first successful demonstration of element specific x-ray detected EPR. This approach will be very valuable for experiments on a number of diluted paramagnetic centers of single molecular magnets or molecular qubits for instance. Such experiments will help to better understand the underlying quantum-coherent states of these highly interesting quantum systems.

Therefore, I recommend publication of this manuscript in Nature Communications as first experimental realization of element specific XDEPR.

We thank the Reviewer for her/his time and efforts. We are very delighted about the entirely positive recommendation. However, we think that our work provides a lot of new information on the technique that goes beyond a proof-of-principle demonstration due to methodological and instrumentation efforts. Essentially, our work establishes a connection between X-ray induced sample charging effects and the localized spin's relaxation times and spin-selective spectra. At the beginning of the study, we did not expect charging effects to be as influential as our results suggest. We also believe that these X-ray related effects are the reason why previous attempts of XDEPR were not successful. By clarifying the relevance of the secondary electrons for XDEPR, our work provides an important foundation to further develop this powerful technique.

Point-by-point response to the comments of the Reviewers. Color code: Reviewers: black normal; authors: blue italic.

Reviewer #1:

I thank the authors for having taken into consideration my comments and improved the manuscript.

I am in favor of its publication overall. However, it is clear from the figure 2b and 2c that only having the computed EPR spectrum it is possible to interpret the spectra. In these conditions, the technique has limited added value to investigate thin films of molecular spins. The authors should consider adding a clear statement about the need to increase the technique's sensitivity and possibly propose a roadmap for it.

We are grateful for the favorable recommendation by the Reviewer. Following the Reviewer's propositions, we have extended the outlook statement in the last paragraph of the manuscript and included possible next steps to enhance the sensitivity.

I have read the explanation they provide about the appearance of a $g=2$ feature in the Cu-edge XMCD detected EPR. The model sounds fine, but if I am not wrong, the simulation in SI refers to the EPR signal. I have not seen the equivalent simulation in the change of magnetization of the Cu^{2+} spin, which is actually what is measured.

I might be wrong here, but I am afraid that most readers will miss this point.

We are grateful for this input to improve the clarity of the SI section. Indeed, we use difference density matrices to model the XDEPR spectra directly at the level of the spin projections, as it is also frequently done in other simulations of EPR spectra. Note that our model does simulate the changes of the longitudinal component $\langle S_{1,z} \rangle$ of the Cu spin S_1 . By contrast, to obtain the EPR spectrum one would calculate the net transverse component of both spins summed together, eg $\langle S_{1,x} + S_{2,x} \rangle$. For more clarity, we have revised the first paragraph such that the longitudinal magnetic moment $\langle \mu_{1,z} \rangle$ is explicitly mentioned.